# Identification and Characterization of a Rare Exon 22 Duplication in *CFTR* in Two Families

**DOI:** 10.3390/ijms26104487

**Published:** 2025-05-08

**Authors:** Simone Ahting, Constance Henn, Maike vom Hove, Vincent Strehlow, Patricia Duffek, Sophie Behrendt, Stephan Drukewitz, Jasmin Berger, Simon Y. Graeber, Julia Hentschel

**Affiliations:** 1Institute of Human Genetics, University Hospital Leipzig, 04103 Leipzig, Germany; simone.ahting@medizin.uni-leipzig.de (S.A.); vincent.strehlow@medizin.uni-leipzig.de (V.S.); patricia.duffek@medizin.uni-leipzig.de (P.D.); sophie.behrendt@medizin.uni-leipzig.de (S.B.); stephan.drukewitz@medizin.uni-leipzig.de (S.D.); 2Pediatricians Office Dr. Henn, Pediatric Pulmonology, 04177 Leipzig, Germany; info@kinderarzt-dr-henn.de; 3Division of Pediatric Pulmonology and Allergology, Hospital for Children and Adolescents, University Medical Center Leipzig, 04103 Leipzig, Germany; maike.hove@medizin.uni-leipzig.de; 4Department of Pediatric Respiratory Medicine, Immunology and Critical Care Medicine, Charité–Universitätsmedizin Berlin, Corporate Member of Freie Universität Berlin and Humboldt-Universität zu Berlin, 13353 Berlin, Germany; jasmin.berger@charite.de (J.B.); simon.graeber@charite.de (S.Y.G.); 5German Center for Lung Research, Associated Partner Site, 13353 Berlin, Germany

**Keywords:** NGS, cystic fibrosis, RNA sequencing, genetic diagnostics, breakpoint analysis

## Abstract

Accurate genetic diagnosis is essential for appropriate treatment in cystic fibrosis (CF). Large copy number variants like duplications in the *CFTR* gene are rare and often classified as variants of uncertain significance (VUSs) due to unknown characteristics of the inserted material, complicating diagnosis and treatment decisions. We identified a previously uncharacterized exon 22 duplication (CFTRdup22) in the *CFTR* gene in two anamnestically unrelated people with CF, both exhibiting a mild phenotype. Initial classification as a VUS was based on standard genetic testing. We employed a custom next-generation sequencing (NGS) panel to determine the exact breakpoints of the duplication and conducted mRNA sequencing to confirm its effect on splicing. DNA and RNA analyses allowed for precise breakpoint determination, confirming that the duplication was in tandem and the reading frame remained intact. This, as well as a residual CFTRdup22 function of ~30% as measured via intestinal current measurement, is consistent with a clinically milder CF phenotype. Collectively, the precise characterization of the variants’ breakpoints, localization and orientation enabled us to reclassify the variant as likely pathogenic. This study highlights the importance of advanced genetic techniques, such as NGS and breakpoint analysis, in accurately identifying CF-causing variants. It underscores the importance of a comprehensive approach and persistence when suspecting a specific genetic condition. This can aid in reclassifying VUSs, providing a definitive diagnosis for the affected family and enabling appropriate therapeutic interventions, including the use of CFTR modulators.

## 1. Introduction

Classifying intragenic duplications according to the criteria set out by the American College of Medical Genetics (ACMG) is challenging due to the unknown location or orientation of the inserted material, aggravating the assessment of its impact on the sequence, reading frame and splicing. Although a majority of duplications are reported to be in tandem [1], newly detected duplications using panel or whole-exome sequencing are often classified as variants of uncertain significance (VUSs) due to a lack of additional lines of evidence. Breakpoint analysis can help bridge this gap, enhancing our understanding of a variant’s impact on DNA and/or RNA levels and providing insights into possible disease mechanisms. Indeed, a 2019 study found that breakpoint analysis for previously uncertain duplications reduced the VUS classification rate from 91 to 35% in breast cancer genes [2]. Hence, analyzing exact breakpoints can aid in improving diagnostic yield and offer families a definitive diagnosis.

In cystic fibrosis (CF), large copy number variants (CNVs) such as deletions and duplications account for 8.7% (102 out of 1167 (25/09/2024)) of all *CFTR* variants recorded in the CFTR2 database (https://cftr2.org/mutations_history (accessed on 22 April 2025)) and for 1.4% (76 out of 5422 of all *CFTR* variants in ClinVar (https://www.ncbi.nlm.nih.gov/clinvar (accessed on 12 March 2025)). Duplications alone represent 1.97% (23/1167) and 0.11% (6/5422) of all variants in CFTR2 and ClinVar, respectively. However, CNVs overall constitute only 1.14% of all *CFTR* alleles in CFTR2, with duplications specifically accounting for just 0.05% (allele number not recorded in ClinVar). Sufficiently rare and often not included in routine screenings, duplications in *CFTR* are therefore frequently underdiagnosed [3], adding to the abovementioned difficulties of proper classification. However, with the advent of genotype-specific modulator therapy in 2020, knowledge of one’s individual genotype is becoming increasingly crucial. Therefore, we previously reported our use of a custom next-generation-sequencing (NGS) panel covering the entire *CFTR* locus [4], enabling the detection of rare and ultra-rare *CFTR* variants in intronic and regulatory regions, as well as the precise identification of large intragenic rearrangements, such as deletions and duplications, thereby demonstrating an overall success rate of 61.7%. This enables not only detection but also more detailed characterization of rare CNVs and provides affected families with definitive diagnoses, potentially opening access to new therapeutic options.

Using this approach, we identified two anamnestically unrelated individuals with the same previously uncharacterized duplication of exon 22 (new numbering) in the *CFTR* gene. Through breakpoint analysis, we precisely determined the extent of the duplication and its tandem arrangement and confirmed the diagnosis of CF in both affected probands.

## 2. Results

### 2.1. Clinical Description

Individual #1: The first individual (#1) is a 3-year-old girl who was diagnosed with CF through a positive newborn screening with an initial sweat chloride concentration of 76 mmol/L and does not have a family history of CF. We identified two *CFTR* variants: the well-established pathogenic maternally inherited variant NM_000492.4(*CFTR*):c.1521_1523del, p.(Phe508del), F508del, and a paternally inherited variant of unknown significance, a duplication of exon 22 (exon 19 in legacy numbering), henceforth termed CFTRdup22. Intestinal current measurement (ICM) showed a reduced CFTR function in the CF range. However, a residual CFTR function was identified in the rectal epithelium with a cAMP response of 47.8 µA/cm^2^ and a total chloride response of 126.5 µA/cm^2^ corresponding to 34% and 29% of normal CFTR activity, respectively [5].

Clinically, she shows occasional dry cough and experiences infrequent lung infections that respond well to antibiotics. A *Pseudomonas aeruginosa* infection shortly after birth was successfully eradicated. Recent swab tests revealed *Staphylococcus aureus*, as well as *Haemophilus influenzae* colonization, both of which were not detected at her last visit at the CF center. Initially, her pancreatic elastase levels indicated a possible onset of exocrine pancreatic insufficiency, but this has stabilized, and her pancreatic function is currently intact. She has no significant gastrointestinal issues and is developing normally. At 3.1 years of age, her height was 96.0 cm (SDS: −0.18/P42.7), her weight was 15.20 kg (SDS: 0.37/P64.3), and her BMI was 16.49 kg/m^2^ (SDS: 0.65/P74.3). Multiple breath washout showed normal lung ventilation homogeneity with a lung clearance index (LCI) of 5.5.

Her current treatment regimen includes daily vitamins, bronchodilator inhalation and hypertonic saline solution inhalation. She started with elexacaftor/tezacaftor/ivacaftor (ETI) triple therapy treatment in late July 2024. At her most recent visit to our CF center, she had been receiving ETI for almost 3 months. She presented in excellent general condition. Her weight had been steadily increasing, and she had not experienced any significant adverse events while on ETI. Additionally, her sweat chloride concentration decreased to 23 mmol/L, indicating a strong response to the treatment.

Individual #2: The second individual (#2) is a 4-year-old girl that was diagnosed with CF shortly after birth based on a positive newborn screening and had an initial sweat chloride test showing 66 mmol/L. At 2 years old, her sweat chloride concentration was 81 mmol/L. We identified a heterozygous maternally inherited variant NM_000492.4(*CFTR*):c.3484C>T, p.(Arg1162*), R1162X, and a paternally inherited duplication of exon 22 (CFTRdup22). Her family history was negative, and it was reported that a younger sibling tested negative for CF at an external laboratory.

She has a relatively mild disease progression, with moderate coughing day and night and a history of infections with *Haemophilus parahaemolyticus*, *Escherichia coli* and *Haemophilus parainfluenzae*, though she currently shows no chronic colonization of CF-associated pathogens. She is pancreatic sufficient but tends toward constipation. At her last visit at our CF center at 3.1 years of age, her height was 96.0 cm (SDS: −0.18/P42.9), her weight was 13.60 kg (SDS: −0.47/P32.0), her BMI was 14.76 kg/m^2^ (SDS: −0.55/P28.9), and she was developing normally. Her current treatments include Ursodeoxycholic acid to prevent liver disease, macrogol laxative for constipation, daily vitamins and inhalations with short-acting beta-agonists and hypertonic saline solution. Unfortunately, as current modulators require at least one F508del allele, she is not eligible for modulator therapy at this time.

### 2.2. Molecular Analysis

The commonly used kit Elucigene CF-EU2v1 for the detection of the 50 most frequent variants in the European population identified the heterozygous pathogenic variants NM_000492.4(CFTR):c.1521_1523del, p.(Phe508del), legacy name F508del (Exon 11), in individual #1 and NM_000492.4(CFTR):c.3484C>T, p.(Arg1162*), legacy name R1162X (Exon 22), in individual #2 (Figure 1A). Whole gene Sanger sequencing of all 27 exons of the *CFTR* gene including flanking regions confirmed these variants (Figure 1B, primer of relevant regions; see Appendix A) but failed to identify a second causative variant. Using MLPA analysis, a duplication of exon 22 of the *CFTR* gene was identified in both girls (Figure 1C), which was named CFTRdup22 (corresponding to exon 19 using legacy numbering). Since MLPA does not allow for breakpoint detection, this variant was at first reported as NM_000492.4: c.(3468−1 _3469+1)_(3717+1 _3718−1)dup, p.? and classified as a variant of unknown significance (VUS, ACMG criteria applied: PM2_supporting, PM3, PP4) due to only one report of an individual with CF in the literature [6] and a lack of functional or RNA studies on the variants’ effect and breakpoints as well as on the reading frame.

To further characterize the duplication, we conducted whole *CFTR* locus sequencing using our previously described [4] custom NGS panel covering the entire *CFTR* locus. Using this panel, we were able to precisely identify the breakpoints of the duplication, confirming its localization in tandem on the DNA level (Figure 2(A#1,A#2)), reporting it as LRG_663:c.3469-52_3717+5032dup, p.?. Still, lacking RNA studies, we were not able to predict this variant’s effect on splicing, preventing us from changing its ACMG classification. Therefore, we collected rectal biopsy tissue and nasal swabs from individual #1 as well as nasal swabs from the father of individual #2, isolated mRNA and subjected it to *CFTR* mRNA specific Sanger sequencing of exon 22 (Figure 2(B#1,B#2), primer; see Appendix A). The results indicated an additional fully spliced exon 22 to be in direct orientation and directly located between exons 22 and 23, creating an mRNA with two adjacent exons 22 and an intact reading frame, most likely not subjected to nonsense-mediated mRNA decay (schematic, see Figure 2C). To visualize the potential structural effects of the duplication, we performed in silico predictions using AlphaFold [7] for both wild-type CFTR and the CFTRdup22 variant (Appendix A). The results suggest alterations in the overall channel conformation, with predicted template modeling scores of 0.74 for the wild-type and 0.70 for CFTRdup22, indicating that the predicted folds are likely close to the true structures. These results, together with the clinically mild phenotype of the individuals, enabled us to finally classify the variant as likely pathogenic according to ACMG criteria (additional criterion PM4 applied), probably belonging to *CFTR* mutation class IV–VI.

## 3. Discussion

Receiving an accurate disease diagnosis is crucial for access to appropriate treatment and better long-term care planning. In CF, a conclusive diagnosis is often mandatory for treatment with genotype-specific modulators such as ETI. A lack of diagnosis or a false result, often arising from a lack of identification or classification of variants as VUSs, can result in complications such as denied treatment access and prolonged reasoning with health care providers for the cost coverage of expensive therapies. Therefore, proper identification and classification of genetic variants are crucial, and further functional assays are often necessary to conclusively interpret variant significance.

Here, we describe the cases of two young individuals, both with mild symptoms of CF, who received inconclusive genetic diagnoses shortly after birth due to the identification of a duplication initially classified as a VUS, due to a lack of further information regarding orientation, localization and breakpoints. Using both our previously described whole-*CFTR*-locus NGS panel [4] to identify breakpoints and subsequent mRNA sequencing for the confirmation of in-tandem localization, we were able to characterize and reclassify the duplication as likely pathogenic, providing both families with conclusive diagnoses of CF. This approach highlights the importance of thorough investigation and underlines the benefits of genomic sequencing as well as breakpoint analysis for large copy number variants.

The reclassified variant, CFTRdup22, leads to the inclusion of an additional, fully spliced exon 22 in the CFTR mRNA. This exon, which encodes 83 amino acids, is duplicated in frame, likely not inducing nonsense-mediated mRNA decay but instead elongating the cytoplasmic nucleotide-binding domain 2 (NBD2). NBD2, alongside NBD1, two transmembrane domains and a regulatory region, forms the chloride channel CFTR. Phosphorylation of the regulatory region, ATP hydrolysis at the NBDs and the dimerization of the NBDs control the channel’s gating activity (reviewed in [8]). NBD1, primarily due to the presence of F508del, has been extensively studied, and variants found in NBD1, like F508del, have been linked to CFTR destabilization, while other NBD1 variants, such as G551D, are associated with impaired channel gating [9]. Similarly, variants located within NBD2 are either known to cause destabilization of CFTR (N1303K, S1235R) or considered to hamper gating (e.g., G1244E, S1251N, S1255P and G1349D) [9,10]. Given the rarity of CNVs in general and the lack of literature on the effects of in-frame tandem duplications, particularly those elongating the NBD2 domain, drawing conclusions about the impact of the duplicated exon 22 on the protein and its function is highly challenging. First in silico predictions using AlphaFold [7] suggest a structural alteration of the CFTR channel that may impair protein function or stability. However, this is unlikely to result in a complete loss of function, as indicated by ICM measurements demonstrating approximately 30% residual channel activity. Furthermore, since the initial case report of a 30-year-old woman with elevated sweat chloride, normal lung function and recurrent pancreatitis [6], the duplication of exon 22 has been described in two additional cases: siblings with F508del on the other allele, both of whom presented with pseudo-Bartter syndrome, though only one exhibited additional steatorrhea, and the other had sweat chloride levels >80 mmol/L [11]. In conjunction with our two patients, who presented with milder CF despite sweat chloride levels around 80 mmol/L, we propose that this duplication leads to a mild CF phenotype when combined with another CF-causing variant on the opposite allele.

These results support CFTRdup22 as a variant likely falling within mutation classes IV to VI, indicative of residual function. These characteristics align with a milder form of CF, characterised by lower sweat chloride concentrations, exocrine pancreatic sufficiency, later age at diagnosis and less severe impairment of lung function [12]. With the recent approval of ETI for all people with CF carrying at least one non-class I variant in early April 2025, it is yet to be determined whether the exon 22 duplication is ETI responsive. This possibility could be explored if individual #2 were to receive ETI, as any observed clinical benefit would be attributable to the CFTRdup22 allele.

Collectively, the data support CFTRdup22 to be a residual function variant, likely belonging to mutation classes IV–VI. It can be classified as likely pathogenic based on ACMG criteria PM2_supporting, PM3, PM4 and PP4. These findings highlight the importance of comprehensive variant characterization—including both DNA and RNA analyses—within the diagnostic process, ideally employing state-of-the-art methodologies. Persistence is particularly crucial when pathogenicity is suspected, especially for CNVs such as duplications, which are often under-characterized but can be reliably assessed with appropriate approaches.

## 4. Materials and Methods

### 4.1. Ethics and Consent

This study adheres to the principles set out in the Declaration of Helsinki. The authors received and archived informed consent of the affected individuals and/or their legal guardians prior to all conducted analyses as well as for the publication of genetic and clinical data.

### 4.2. CF-EU50

Multiplex PCR amplification and subsequent capillary electrophoresis were carried out using the CF-Eu2v1 (Elucigene^®^, Delta Diagnostics, Manchester, UK) assay for the detection of the 50 most common pathogenic variants in the *CFTR* gene.

### 4.3. Sanger/MLPA

Sanger sequencing was performed on all 27 exons and the flanking intronic regions using the Applied Biosystems (Waltham, MA, USA) 3500 Genetic Analyzer. MLPA analysis was carried out with the P091-D1 kit (MRC-Holland) following the manufacturer’s protocol. The results of both Sanger sequencing and MLPA were analyzed with Sequence Pilot Software v.5.4.2 (JSI Medical Systems, Ettenheim, Germany).

### 4.4. NGS

Next-generation sequencing of both probands and their parents for the entire *CFTR* locus was conducted and interpreted as previously described [4]. Breakpoints were identified via visual inspection using the Integrative Genomics Viewer (IGV) [13]. Written reports were sent out to the clinicians. Variant classifications were uploaded to ClinVar [14].

### 4.5. Variant Interpretation

All variants were described in regard to GRCh38 (NM_000492.4) and classified according to the ACMG criteria [15]. Wherever possible, systematic numbering was used for the specification of exon numbers. The following databases were used for variant classification: ClinVar [14], HGMD [16], CFTR2 [17], CFTR-France [18], gnomAD v4.1.0 [19], REVEL [20] and SpliceAI-lookup [21].

### 4.6. RNA Analyses

Since CFTR is not expressed in whole blood, nasal epithelial cells were collected from the nasal epithelium by brushing the nasopharyngeal cavity through both nostrils. The swab was then placed into a tube containing 1 mL of RNAlater™ Stabilization Solution (Thermo Fisher Scientific, Waltham, MA, USA), and the brush was cut off at the head. The swab in the solution was subsequently transported at room temperature to the laboratory. Upon arrival, the swab was gently rotated and rubbed along the inner wall of the tube several times to ensure maximum release of cells into the solution. Alternatively, rectal biopsy tissue was inserted into RNAlater™ Stabilization Solution and snap-frozen, then sent to the laboratory on dry ice. Once in the lab, the tissue was homogenized using a one-time-use plastic pestle in 1.5 mL tubes. The cell-containing solutions from both nasal swabs and rectal tissues were then centrifuged at 15,000× *g* for 5 min at room temperature. The supernatant was discarded, and RNA was isolated using the RNeasy Mini Kit (Qiagen, Hilden, Germany) according to the manufacturer’s instructions, with the following modifications: cell lysis was performed using RLT buffer supplemented with 1% (*v*/*v*) β-mercaptoethanol, DNase digestion was extended to 30 min (instead of 15 min), and RNA was eluted in 30 μl RNase-free water.

Complementary DNA (cDNA) was synthesized using the PrimeScript™ RT Master Mix (TaKaRa Bio Europe SAS, Saint-Germain-en-Laye, France) according to the manufacturer’s instructions. Reverse-transcription PCR (RT-PCR) was performed (see Appendix A for primer sequences), and the PCR product was purified from an agarose gel using the QIAquick Gel Extraction Kit (Qiagen, Hilden, Germany). The purified Exon 22 CFTR mRNA PCR products were then sequenced using the same primer pairs employed in the RT-PCR, utilizing the Applied Biosystems 3500 Genetic Analyzer (Thermo Fisher Scientific, Waltham, MA, USA). Bioinformatic analysis, including sequence alignment and identification of duplicated regions, was carried out using IGV, in conjunction with sequence data obtained using the UCSC Genome Browser [22].

### 4.7. Intestinal Current Measurement (ICM)

ICM was performed as previously described [5,23,24]. In brief, superficial biopsies of the rectal mucosa were collected by endoscopic forceps biopsy and mounted in perfused micro-Ussing chambers. CFTR function was assessed as cAMP response (100 µmol/L 3-isobutyl-1-methylxanthine (IBMX) and 1 µmol/L forskolin) and total chloride response (IBMX/forskolin and 100 µmol/L carbachol). The percentage of normal CFTR function was calculated using age-dependent reference values [5].

### 4.8. Multiple Breath Washout (MBW)

MBW testing was performed with the Exhalyzer D system (Eco Medics, Duernten, Switzerland), 100% oxygen was used to wash out resident nitrogen from the lungs, and the measurement was evaluated using spiroware 3.3.1 (Eco Medics, Duernten, Switzerland) [25,26,27]. The upper limit of normal (ULN) was determined as 7.1 [28].

## Figures and Tables

**Figure 1 ijms-26-04487-f001:**
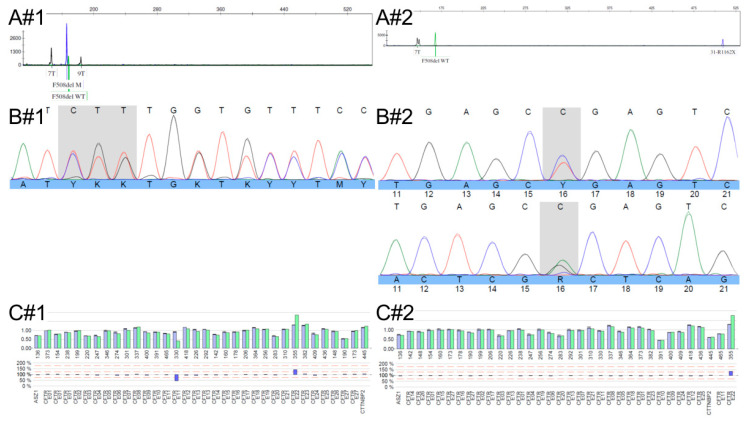
DNA analysis results for individuals #1 and #2. (**A**) Results from the Elucigene CF-EU2v1 assay, where blue peaks indicate the presence of variants, and green peaks represent wild-type alleles; both F508del (individual #1) and R1162X (individual #2) variants were detected (controls were inconspicuous); (**B**) Sanger sequencing results (only the forward primer was usable for individual #1, as shown), confirming the detected variants; peaks represent nucleotides in the DNA sequence, where each nucleotide has a different colour; A = green, T = red, C = blue and G = black. (**C**) MLPA results showing all 27 *CFTR* exons along with flanking probes of adjacent genes; control data in blue, patient data in green; blue bars at the bottom indicate the loss or gain of probes in the MLPA (loss of Exon 11 in individual #1 is attributed to probe binding interference caused by the F508del variant on the other allele).

**Figure 2 ijms-26-04487-f002:**
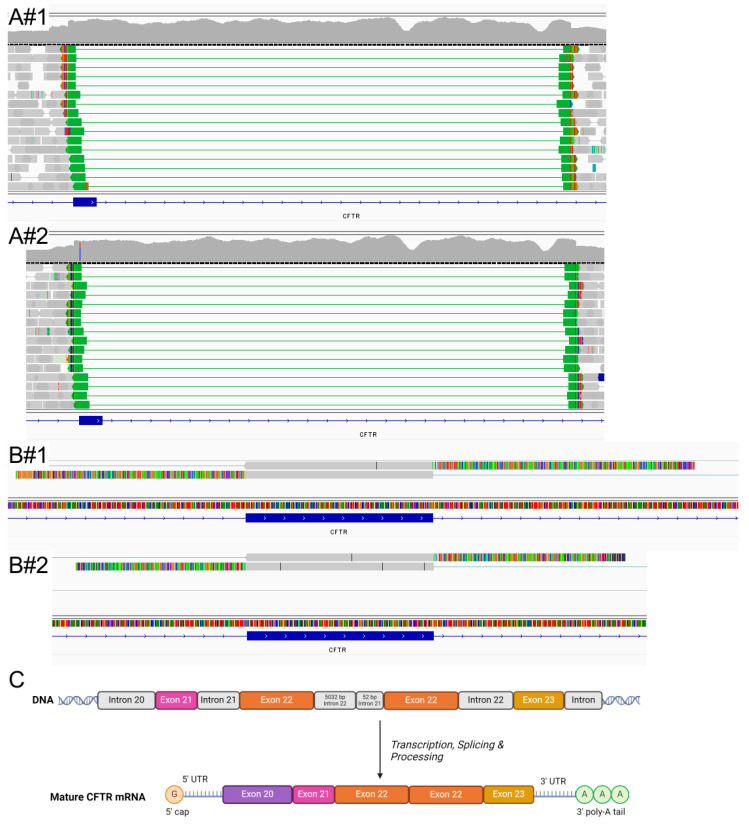
Further characterization of CFTRdup22 in individuals #1 and #2. (**A**) Integrative Genomics Viewer (IGV) view of DNA short-read NGS panel results; on top the coverage track, on bottom the sequencing reads; shown are the duplication of exon 22, breakpoints, increased coverage and predicted tandem localization (green reads); in individual #2, the R1162X variant in exon 22 is also visible (see coverage track for variant call (vertical line indicates variant)). (**B**) Sanger RNA sequencing, viewed in IGV, for exon 22 with primers in exons 21 and 23. Forward (top) and reverse (bottom) reads for the rectal biopsy tissue of individual #1 and nasal mucosa swab of the father of individual #2 show soft-clipped regions (colorful part of the read) on both sides of exon 22, confirming the presence of adjacent, non-aligned sequences. Sequence analysis of these soft clips confirms exon 22 duplication. (**C**) Schematic representation of exon 22 duplication on the DNA and RNA level: on DNA, exon 22 is followed by 5032 bp of intron 22, 52 bp of intron 21, another exon 22 in direct orientation, before a full intron 22; on RNA, the in-frame duplication shows exon 22 repeated in tandem; for simplicity, only exons 20–23 are shown (created in BioRender.com).

## Data Availability

The data presented in this study are available on request from the corresponding author due to privacy protection of the individuals described in this research.

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
