# Peer review of "Identification and Characterization of a Rare Exon 22 Duplication in *CFTR* in Two Families"

_ijms, 2025, doi:10.3390/ijms26104487_

Round 1

Reviewer 1 Report

Comments and Suggestions for Authors

This is a well-written, well-organized, and rigorous report that describes the identification of a rare, tandem duplication of CFTR exon 22 in two unrelated CF pediatric patients. The authors utilize a variety of methodologies to identify this rare mutation that includes multiple mutation detection methods as well as mRNA analysis, and its effect on CFTR function. The results help to verify the CF diagnosis, identify a rare mutation, and aid in whether the patients would potentially benefit from  CFTR modulator therapies. The methods are well-described and their interpretation is logical and well-supported by multiple, independent methods. I do not have any criticisms of this manuscript. I believe it is ready to be accepted. 

Reviewer 2 Report

Comments and Suggestions for Authors

General comments for the authors:

Thank you for providing the detailed report of these two CF patients that carry CFTR exon 22 duplications. Presentation of results is concise and well structured. This report is a valuable contribution to the field. Two minor comments below.

Specific comments:

Lines 49 - 53 –  I get slightly different numbers for CFTR2 (25/09/2024 version); the biggest difference is for alleles in CFTR2 that are duplications, I get 0.05% alleles (110 duplications vs 211,106 total alleles). Other differences are minor – I get 99 CNVs and 23 duplications.  

Line 223 – Just need to clarify if the kit screens for 50 or 51 variants. On line 117 it says “…50 most frequent variants…” on this line it says “…51 most common…”.

Reviewer 3 Report

Comments and Suggestions for Authors

The presented case report describes the identification and genetic diagnosis of a rare CFTR mutation. As stated by the authors, large segment duplications or deletions in genetic sequences can be challenging to diagnose, and by the use of conventional genetic testing, these mutations mostly remain unrecognized. This study describes the use of an advanced NGS method developed by the same researcher group earlier. This methodology has the capability to identify breakage points and surface the genetic context at the site of such a mutation.   By the use of the developed method, the authors provide evidence that the exon 22 duplication is present on one of the CFTR alleles in tandem in both patients. The duplicated sequence encompasses the entire exon 22 sequence, along with a partial intron portion. Accordingly, the coding sequence remains in frame on the level of mature mRNA. Therefore, the full length of the CFTR protein is synthesized from the mutant transcript that consists of an extra peptide segment in tandem at the NBD2 CFTR domain. In vitro functional test shows that the mutated protein from this allele retains residual function, which secures pancreatic sufficiency and intermediate sweat chloride levels, resulting in a milder CF phenotype. The manuscript is well written, and the conclusions are supported by the results. The references are adequate and support the manuscript.

The following minor changes are suggested:

  • At line 50, please reference the CFTR2 website as https://cftr2.org/mutations_history.
  • At the end of line 50, please reference the ClinVar website as https://www.ncbi.nlm.nih.gov/clinvar.
  • Please rephrase the sentence at lines 185-187: “This exon, which encodes 83 amino acids, is duplicated in frame, likely preventing nonsense-mediated mRNA decay and instead elongating the cytoplasmic nucleotide-binding domain 2 (NBD2).” Most likely, this in-frame duplication is not recognized by NMD because it does not present an early termination codon and subsequent exon-junction complex remnants on mature CFTR mRNA that can induce this quality control mechanism. So, the duplication is not preventing, but rather does not induce NMD because of the in-frame insertion.
  • Please rephrase the sentence at lines 212-214: “Moreover, since individual #1 also carries F508del on her second allele and qualifies for ETI treatment, her clinical improvements suggest that CFTRdup22 responds well to the triple modulator therapy ETI.” ETI is particularly effective and benefits patients carrying F508del CFTR protein. Correcting this variant by ETI has been shown to be highly beneficial for patients carrying only one copy of F508del variant on one of the CFTR alleles with another disease-causing variant (none F508del) on the second allele. The effectiveness of ETI treatment for the mutation CFTRdup22 could be proven by performing an n-of-1 trial involving patient #2 (not #1), where the second allele (R1162X) represents an unresponsive CFTR variant toward ETI modulation.  Therefore, any benefit of modulator treatment that surfaces during the clinical trial would be attributable to the protein arising from the CFTRdup22 allele.

Reviewer 4 Report

Comments and Suggestions for Authors

In case report: Identification and Characterization of a Rare Exon 22 Duplication in CFTR in Two Families, the authors presented two children with Cystic fibrosis which are carriers of mutations in CFTR gene including F508del and R1162X. In both children as a second mutation in CFTR gene, duplication of exon 22 is detected, CFTR dup22. The CFTR dup22 variant is currently classified as variant of uncertain significance, while the authors suggest its reclassification.

Since in this case report the duplication CFTR dup22 is suggested to be reclassified as disease causing variant, which have significant impact on identifying newborns/individuals with Cystic fibrosis the report is worth of publishing.  

Major and minor suggestions for authors:

  1. The structure and content of Abstract need to be reevaluated, please check the instructions for authors, usually the journal IJMS does not require structured abstract.

Line 27, Both individuals exhibited mild phenotype, this is not result of analysis, they already had elements of CF diagnosis and that is why the second mutations were searched, it is the description of subjects’ pathology.  

State clearly how you suggest the duplication to be reclassified in abstract, according to your study, instead of likely pathogenic (L27).

  1. Line 76 avoid using double brackets.

  1. If not provided, please provide Figures 1 and 2 in color.

  1. Since the authors have determined the sequence of CFTR gene harboring the CFTR dup22 variant, it is strongly suggested to provide a 3D model of CFTR protein with this variant and to present the Figure with wt and mutated protein, by using some of bioinformatics tools.

  1. Last paragraph in Discussion needs revision. Please clearly state what is the conclusion of your Case report, and how you suggest CFTR dup22 to be classified, in order to emphasize the novelty of your study.

  1. Explain in Discussion why CFTR dup22 was not previously recognized as disease causing mutation, which is what you want to correct.

Round 2

Reviewer 4 Report

Comments and Suggestions for Authors

Authors need to make corrections in Abstract:

people with CF (pwCF), is not commonly in use, please remove abb., it is patients with CF

(ACMG criteria: PM2_supporting, PM3, PM4, PP4) this is unnecessary for abstract

last 2 sentences in abstract need to be rewritten to explain want is to novelty of your study and what new it brings.

for example ….in accurately diagnosing CF caused by rare CFTR variants

in identifying CF causing variants.

Last sentence in abstract, emphasize the novelty of your report.

This is consistent with a residual CFTR function of ~30% - specify what exactly is consistent with 30% of CFTR function … that allows the reclassification of the variant as likely pathogenic

If not repeated throughout the text there is no need for pTM abbreviation, also abbreviation needs to be define every first time it appears in text, abstract and Figure legend, please check the Instruction for authors. It relates to all manuscript text.

 Intestinal current measurement (ICM) once the abbreviation introduced in text do not need to repeat, this relates to all Abb in the manuscript, check instructions.

please change classifiable into for example which can be classified as…..
